# Use of a Novel Resistive Strain Sensor Approach in an Experimental and Theoretical Study Concerning Large Spherical Storage Tank Structure Behavior During Its Operational Life and Pressure Tests

**DOI:** 10.3390/s20020525

**Published:** 2020-01-17

**Authors:** Virgil Florescu, Stefan Mocanu, Laurentiu Rece, Robert Ursache, Nicolae Goga, Constantin Viorel Marian

**Affiliations:** 1Technological Equipment Faculty, Mechanical Technology Department, Technical University of Civil Engineering of Bucharest, RO-020396 Bucharest, Romania; mocanustef@gmail.com (S.M.); rece@utcb.ro (L.R.); robionut@gmail.com (R.U.); 2Foreign Languages Engineering Department, University Politehnica Bucharest, RO-060042 Bucharest, Romania; n.goga@rug.nl (N.G.); constantinvmarian@gmail.com (C.V.M.)

**Keywords:** resistive strain sensor, large pressurized spherical tank, FEM, structure optimization

## Abstract

This paper introduces a new method for the use of tensor-resistive sensors in large spherical storage tank equipment (over 12,000-mm diameters). We did an experiment with 19 petroleum or ammonia product sphere-shaped storage tanks with volumes of 1000 and 1800 cubic meters, respectively. The existing literature only contains experiments based on sensors for tanks with diameters no larger than 600 mm. Based on a number of resistive strain sensor measurements on large spherical pressurized vessels regarding structural integrity assessment, the present paper is focused on the comparison between "real-life" obtained sensor data versus finite element method (FEM) simulation results. The present paper is structured in three parts and examines innovative directions: the use of the classic tensor-resistive sensors in a new approach concerning large structural equipment; an original 3D modeling method with the help of the FEM; and conclusions with possible implications on the regulations, design, or maintenance as a result of the attempt of mutual validation of the new methods previously mentioned.

## 1. Introduction

Experimental structural strain-field assessment by means of a resistive strain sensor approach and finite element method (FEM)-based optimization concerning the geometrical positioning of resistive strain sensors are the subject of many research studies [1,2,3,4,5,6,7,8,9,10]. Moreover, the study of large, spherical, storage-specific tank structures becomes a real challenge due to geometric particularities. To the best of our knowledge, the existing literature contains only experiments based on sensors for tank equipment with diameters no larger than 600 mm. Liquid petroleum gas (LPG)- and ammonia-filled spherical tanks (used for our sensors measurements) are critical equipment. Their failure (cracks, explosions, etc.) can lead to massive losses in terms of human life and goods.

In practice, such tanks are periodically tested while functioning to check their operating condition. The investigation and assessment of current equipment are performed for safety purposes. In accordance with the law, the pressure vessels are subjected for periodically inspection. A test consists of checking whether the thin wall of the vessel is being subjected to an overpressure (about 1.25 times the maximum authorized operating pressure) and remains in the elastic domain [11,12,13,14]. Because the test procedure requires the measurement of the internal pressure at the top of the sphere, due to the large diameter of the structure, the water column additionally strains the mantle in the bottom area with an extra 0.12 to 0.16 MPa; for some cases, this represents an overstrain of about 15–20%. The problem becomes even more complex when one needs to estimate the remaining service life of the installation according to existing regulations. As noted above, such tanks are critical equipment and it is important to test them for proper functioning, especially in the light of newer findings. For example, as reported in the paper, the stairs (which are not nationally or internationally regulated as constructive parts of large pressure vessels) may interfere with the proper functioning of the tanks.

Most of the studies carried out in this field have focused on the pipe-type structures used with measurements performed by means of bidirectional tensor-resistive sensors [2,3], or studies carried out under laboratory conditions [4,5,6,7]. For example, Agbo et al. presented a study [1] that describes investigations performed using tensor-resistive sensors regarding the behavior of some thin-walled metal structures with an operating usage history and subjected to an internal pressure. In comparison with these laboratory studies, and from the point of view of both FEM simulations and the experimental approach, this paper addresses large spherical structures that require in situ treatment and the use of three-direction sensors.

Regarding the way in which the thickness of the wall was treated in the FEM from the meshing point of view, Zhu et al. [5] approached the problem of resistance and stability by applying FEM on some spherical structures; however, they focused only on the manhole area, while the experiment was performed on some laboratory models (experimental verification using acrylonitrile butadiene styrene (ABS) scale models). In the case of the FEM application for small structures (several millimeters in size), the meshing operation does not raise particular problems in terms of the size or shape of the meshing element, alleviating the need for heavy computing resources. For a structure with dimensions of approximately 1.5 mm^2^, Messina et al. [7] used a meshing network with 250,000 elements. As compared with his work (and other similar ones), in our case, considering the physical size of the structure discussed in this paper (diameter between 12,000–16,000 mm with wall thicknesses between 20–45 mm), applying the FEM and obtaining useful and verifiable results proved to be a real challenge. As a starting point, we used the optimization method for single- and multi-axis load cell structures developed by Takezawa et al. [6], which was adapted to the complexity of the structures that were the subject of this study. This also led to the establishment of a new original problem-solving approach that implied the customization of the 3D modeling algorithm consisting of ten slice modules for large structures.

The present study proposed a comparative study of the results obtained by applying the FEM and the results obtained following resistive strain sensor measurements during in situ overpressure tests. The results obtained by applying the FEM according to the original problem-solving approach were useful both for finding the position of the resistive strain sensors for the experimental study, as it has been done in a similar way in References [8,9], and for the theoretical/analytical determinations of the specific values for the state of the tension and deformation (stress/strain field values). What makes our research original is the use of SolidWorks algorithmic symmetry for the FEM 3D modeling and simulations for large structures. To the best of our knowledge, this slicing modeling technique is unheard of in the scientific literature for complex and large structures.

The paper is structured as follows: Section 2 describes the material and methods used, Section 3 provides the results and discussions, and Section 4 shows the conclusions.

## 2. Methods, Materials, and Means of Investigation

The geometrical parameters, the working fluid, and the test pressure value for all 19 sphere tanks that were investigated and analyzed are given in Table 1.

This article presents the results obtained (and their comparison) by using two methods: in situ determination, namely stress assessment by applying resistive strain sensor measurements during overpressure tests under actual loads, and a theoretical one, by means of FEM.

The determination of the tension and deformation state of the strained bodies can be done analytically (by means of analytical calculation and/or numerical methods) or experimentally [15,16,17].

For the theoretical determination of tension and deformation state, the acceptance of simplifying hypotheses is required regarding the shape and structure of the element, the mechanical features of the material the element is made of, and/or its loading and support scheme. Moreover, in the above-mentioned papers, the material of the element on which the calculations are performed is considered ideal: continuous, homogeneous, isotropic, and perfectly elastic. In reality, these conditions are not fully met because of real conditions concerning technological and engineering processes [18,19]. In the case of bodies or elements with a more complicated geometrical shape and loading scheme, analytical calculation with numerical methods is quite difficult and requires the input of a trained operator for this purpose, as well as a prudent use of simplifying hypotheses.

For the FEM, results that are shown below were measured for sphere no. 8 (position 8 in Table 1), which was filled with LPG and had a volume of 1000 m^3^, a working pressure of 1 MPa, and a test pressure of 1.3 MPa. For all other spheres, results are similar.

According to the manufacturing documentation, the spherical tank was made of structural steel with the minimum mechanical features, which were as follows (Table 2), as recorded for one of the ferrules that make up the mantle:

The design is in accordance with EN 13445 [11,12]. Consequently, the determination of the admissible stress (noted f) is defined as the minimum value between the ratios Rc20/1.5 and Rm/2.4, where Rc20 is the tensile strength at 20 °C and Rm is the yield strength in MPa.

As a result of the EN 13445-3 calculus specification, the minimum required thickness for spherical shells depends on nominal design stress (f).

The normal design stress (f) is defined as:

f = min (Rp/1.5; Rm/2.4) where Rp is the yield strength and Rm is the tensile strength.

f = min (380/1.5; 560/2.4) = 233.33 MPa.

This value was also used as an input parameter for the FEM approach. For further reading regarding the FEM mathematical apparatus, we refer the reader to References [1,5,20]. General information regarding the FEM computational approach is given in the Appendix A.

For experimental evaluation, the method involved finding the right resistive strain sensors and the identification of the most appropriate areas to make the measurements from, along with documentation and validation of the most probable perilous areas. To find the right areas, one method was to compute the maximum stressed areas of a structure (in our case a tank) through FEM 3D modeling and simulation.

The authors’ initial purpose was to compare the experimental results measured with the simulated values using the FEM. However, unexpected results were obtained. Anticipating further discussions, our paper reveals the fact that the access stairs do influence the mechanical behavior of the tanks. No regulations at the national or international level consider the effect of the stairs. To the best of our knowledge, this result is not reported in the literature. According to the results, our recommendation is that access stairs should be considered by existing regulations related to the mechanical behavior of the tanks. This is further detailed in the next sections.

## 3. Results and Discussion

### 3.1. FEM Simulations

FEM simulations have both qualitative and quantitative components. The qualitative component consists in the identification of the most stressed areas, which determines the scheme of the sensors’ arrangement. The quantitative component consists in estimating the value field concerning the primary strain analytical calculations.

In the FEM analysis, the following simplifying hypotheses were considered: the thickness of the material was uniform for the entire thin wall of the sphere and the thickness of the wall was the smallest value measured using an ultrasound technique.

The software used for modeling and simulation was Dassault Systèmes SolidWorks 2014 (Watham, MA, USA) [21].

It is a well-known fact that SolidWorks represents a complex software tool for design and simulation purposes; to emphasize this aspect, one can quote J. Ed Akin’s work, *Finite Element Analysis Concepts via SolidWorks, 2009*: SolidWorks “studies calculate displacements, reaction forces, strains, stresses, failure criterion, factor of safety and error estimates. Available loading conditions include point, line, surface and thermal loads, elastic orthotropic materials are also available. The SW Simulation software also offers several types of nonlinear studies” [20].

Application of SolidWorks Simulation 2014 allows for several approaches for 3D modeling and simulation:

(1) A first criterion for the selection of meshing is the use of the “part” type elements (individual parts) or “assembly” (entity made of several individual parts) (see Figure 1 and Figure 2).

(2) Second, a simplification can be made such that the simulation time is shortened and less computational resources are used by using the symmetry option (circular, in this case) when 3D modeling and meshing the element that is going to be studied.

For our mesh tank simulations, by using a single element (using option “part” without the use of symmetry), one can obtain the model in Figure 1a. Some of the existing disadvantages are the impossibility of choosing different materials for the elements, namely the spherical tank and its support legs, respectively. Another disadvantage is the existence of some “residues” (see Figure 1b), as seen on the transversal section of the element from Figure 1a; one can observe the traces of the intersection of the legs of the spherical tank with the tank itself (in the center). We call these traces 3D modeling residues; these residues will cause problems related to the application of the FEM method (with implications for the difficulty of meshing and, implicitly, for the degree of accuracy).

As a first measure for raising the accuracy of the treatment, one can employ 3D modeling with the use of SolidWorks symmetry. This method allows for the choice of several variants, among which is the algorithmic treatment with the help of circular symmetry, which we consider to be of maximum efficiency in our case in terms of the number of meshing elements used and the computational simulation time. The circular symmetry allowed us to use just a slice (Figure 2a) of the tank element for meshing and simulation (as represented in Figure 1c) from a total number of ten slices in which the tank was divided. The input data of the simulation are presented in Figure 2b,c. The results obtained in this case were expected to be closer to the real values because this method allowed fora larger number of finite elements because the modeled slice portion was smaller compared to the one for the whole tank from Figure 1a.

According to SolidWorks representation rules (Figure 2a,b), different red-sized arrows stand for forces or pressure fields and green custom arrows and indicators stand for boundary conditions in terms of degrees of freedom (displacements) as supporting scheme.

In Figure 2b, the loads coming from internal pressure, weight of the contained LPG, and weight of the tank itself are represented. In Figure 2c, one can notice the local coordinate system, denoted Coordinate System1 (Figure 2c), in relation to which the law of linear variation of the load could be modeled according to the actual weight of the LPG stored in the liquid phase by considering an 80% full container (according to the working manual instructions, a tank will be filled only 80% from the total volume, where the rest should remain empty for vaporization phenomena). The FEM parameters are represented in Figure 2c.

We performed simulations with one slice as represented in Figure 2a–c. In order to validate and confirm the results obtained with one slice, we used also the option of “assembly” for meshing and simulation. In this case, we used the whole set of ten slices without using the method of simplifying the calculation scheme by using symmetry (see Figure 2). As noted in Figure 2c and Figure 3c, the mesh grid execution time for one slice with options “part” and “symmetry” was approximately 10 s compared to 1.5 min for an assembly composed often slice modules. Moreover, one can see (Figure 3b) a better 3D modeling quality (as compared to Figure 1b) due to the total absence of residual artifacts.

The use of the SolidWorks symmetry algorithm for the FEM 3D modeling and simulations of the large structures considered makes this work original; to the best of our knowledge, for complex and large structures, this slicing modeling technique is unheard of in the scientific literature.

The results of the FEM simulations (option “assembly”) are: the values of the equivalent strain field (Figure 4a), the displacements components field (Figure 4b), the von Mises equivalent stresses (Figure 4c), and the maximum von Mises equivalent stresses field (Figure 4d). In the mechanics of materials, the von Mises yield criterion (also known as the maximum distortion energy criterion) can be formulated in terms of the von Mises stress or equivalent (von Mises) tensile stress, which represents a computable scalar value of stress; a material is said to start yielding when the von Mises stress reaches a value known as the yield strength. The von Mises stress is used to predict the yielding of materials under complex loading from the results of uniaxial tensile stress. Based on a known plane stress state defined by principal normal stresses $\sigma_{1},\sigma_{2},$ the equivalent von Mises stress is given as Equation (1):(1)$$\sigma_{von\;Mises}=\sqrt{\sigma_{1}^{2}+\sigma_{2}^{2}-\sigma_{1}\sigma_{2}}.$$

### 3.2. Experimental Method

In what follows, we present the results for the experimental method.

Using the FEM simulations, we determined the zones (the area of intersection of tank with supporting legs) where the resistive strain sensors were to be placed. Using the sensors mounted in the determined places, we obtained the experimental data. In this section, we compare the theoretical von Mises equivalent stress obtained using the FEM method with the experimental ones.

The resistive strain sensor electro-resistive measurements method (measuring electrical resistance variations undergone in a strain gauge sensor grid linked to specific resistive strain sensor bridge equipment) represents one of the most frequently used experimental techniques and targets the real behavior of the material of the mechanical structure that is being investigated. The option to apply this method is based on the fact that research can be carried out on the structure under actual operating conditions.

It is well known that for materials situated in the elastic behavior limit (Hooke’s theory), there is a linear relationship between specific deformations (strains) and stresses. Above this limit, plastic deformation occurs and the relation between the specific deformations and stresses is no longer linear; moreover, the equations that express the stress/strain connection become very complex.

Any specific structural element deforms under a load, i.e., the existence of a normal/tangential field stress (σ, τ). Experimental direct calculation of such tensile parameters is impossible; therefore, in order to address the issue, one can experimentally determine the corresponding strain field and then, based on theoretical relationships (Hooke’s theory) between specific deformations and stresses, establish the stress field values.

The main disadvantage of the experimental method resides in the fact that it does not directly identify the most strained areas of the equipment, a situation that may be overcome with the help of the previous FEM study.

Starting from the geometrical and constructive particularities of the tank, and in order to be able to determine the von Mises stress field state, it has been decided that for each measuring point, a three-directional strain sensor (rosette type) should be used (three-directional strain sensors give the strains according to three directions and this can be used to characterize the principal stresses and their directions; other types of sensors are not adequate for this). The location plan of the resistive strain sensor is shown in Figure 5.

In Figure 5, the original constructive design of the tank is presented together with the sensor placements. As observed in the figure, the constructive design is also formed by slices that are not correlated with the ones from the theoretical FEM simulation (the only common part is that both slices include the area of intersection between the sustaining legs and the tank). In Figure 5, it can be observed that only five sensors were used. This was due to regulatory measurements rules for such tanks and due to the consideration of costs. The policy was to first place a smaller number of sensors, and only if abnormal irregularities were observed, more sensors would be deployed on the tank and the measurements would be repeated. In our case, five sensors were considered enough to be used for the first series of measurements. As determined from the FEM simulations sensors were placed in the area of the intersection between the legs and the tank (as seen in Figure 5).

In order to perform load-related strain field assessment, a rosette-type sensor with three measurement directions (Figure 6) was used, with the corresponding temperature characteristics shown in Figure 7. The sensor manufacturer recommends the use of temperature correction curve 2 from Figure 7 for a HBM 6/350 CRY81-3L-3M-type sensor (Hottinger Baldwin Messtechnik Gmbh, Darmstadt, Germany).

### 3.3. Comparison Between the FEM Simulations and Experimental Results

Table 3 indicates the sensor-recorded values (equivalent von Mises stress) with respect to the most strained/stressed point of the supporting legs. The first five columns correspond to the five points from Figure 5, while the sixth column corresponds to the FEM-calculated values. Characteristics of the nineteen tanks are given in Table 1. The calculated FEM stress from column 5 was the one obtained with the “assembly” option. Given the symmetry, the stress was the same for all ten FEM slices; therefore, we put just one value in Table 3. Because we did both with the “assembly” and “part” options, we took the highest stress value from the two FEM simulations to emulate a worst-case scenario; for all cases, although the values were similar, the highest values were given by FEM simulations with the “assembly” option.

From Table 3, it can be observed that FEM values and experimental values for points P2, P4, and P5 were similar, while P1 and P3 were not. For validating this statistically, we used Tukey’s test. Through this test, which verifies the equality of the means for “*k*” selections of possibly different volumes, we aimed to identify the similarities and differences between the values measured experimentally for each supporting leg and the value calculated using the FEM.

The means ${\bar{X}}_{j}$ were calculated using the Equation (2), which determines ${\bar{X}}_{\min}$ and ${\bar{X}}_{\max}$:(2)$${\bar{X}}_{j},j=\bar{1},\bar{k},$$
where “*k*” is the number of different volumes involved and *j* the number of selected measurement points. 

The formula for Tukey’s test is:(3)$$q=\frac{{\bar{X}}_{\max}-{\bar{X}}_{\min}}{SE},$$
where ${\bar{X}}_{\max}$ is the larger of the two means being compared, ${\bar{X}}_{\min}$ is the smaller of the two means being compared, and *SE* is the standard error of the sum of the means, which is calculated using Equation (4):(4)$$SE=\sqrt{\frac{S_{1}^{2}+S_{2}^{2}}{n}},$$
where *S*_1_^2^ and *S*_2_^2^ are the dispersions for the two corresponding selections ${\bar{X}}_{\min}$ and ${\bar{X}}_{\max}$.

Appendix B contains the raw data and auxiliary information.

The computed “*q*” value was compared to a certain value obtained from the standardized range distribution “*q_a_*” for n–k degrees of freedom, where n = 19, the total number of investigated equipment, and k = 2. One can consider *q_a_* to be the critical value. It was considered that the values were different if the computed *q* value was greater than the critical value *q_a_*.

The results are presented in the Table 4.

The values obtained for *q_a_* with 17 degrees of freedom are presented in Table 5.

Using Tukey’s test, it was concluded that P2, P4, and P5 value were consistent with the FEM analysis. A significant difference was observed for the values corresponding to points P1 and P3 in relation to the FEM-calculated value. In all cases, the most stressed point was P1, which corresponded to the position of the leg attached to the tank access stairs. Accordingly, the less stressed point was the diametrically opposed one, which was P3 in all cases. Excluding the most stressed point (P1) and the least stressed point (P3), it can be seen that the values for P3, P4, and P5 measured using the tensor-resistive sensors in three directions were within the margin of ±3% compared to the values calculated using the FEM method. This was quite remarkable and doubly validates both the FEM simulations and the experimental method (one validates another).

The difference between the values recorded by sensors at P1 (adjacent to the tank stair access structure) and the opposite one (P3) was quite significant, with this phenomenon being systematically reported in all cases of the investigated spheres. As noted, P1 was the most stressed point while P3 was the least stressed. The access tank stairs was adjacent to P1, placed between P5 and P1. In this context, P3 can be considered the opposite point to P1 (and not also P4), where the position of the access stairs explains why P3 was the less stressed (and not also P4). One result of the study is the fact that the presence of stairs causes a peak stress value point at P1. Such an unconventional structural behavior was caused by specific interface areas (between the stairs and tank sphere) that are prone to stress concentration due to a sudden local increase in terms of general mechanical stiffness caused by the stair’s presence, with a corresponding undesired influence from the point of view of the mechanics of materials.

Hence, from the point of view of the general stress state field, the sensor data in Table 2 indicate an unacceptable trend concerning the structural fatigue strength (in point P1) with poor life expectancy characteristics.

One can ask why we did not model the stair structures using the FEM. Our starting point for FEM modeling was the European regulatory norms, as defined in EN 13445 (similar in USA and worldwide), which is particular to our tank spheres. This regulation does not include the stairs in the mechanical design of the sphere; therefore, we did not include the stairs in the FEM modeling. However, the experimental results show clearly that the stairs had an influence on the stress behavior of the tank; however this, as noted, is not regulated. We will enlarge this discussion in the next section.

## 4. Conclusions

Based on the observations from the experimental data and the FEM, we argue that there is a need for a new algorithmic convergence from the design, manufacture, and maintenance perspectives. The study revealed areas that, during periodic tests, could reach the flow limit as a result of initial constructive solutions.

Based on the experimental results recorded for the 19 spheres subjected to resistive strain sensor measurements (electrical resistance variations undergone in a rosette-type three-directional sensor grid linked to a specific resistive strain sensor bridge equipment) obtained using periodic technical state evaluation of spherical thin-walled pressure vessels, a notable difference between the values obtained using 3D modeling via finite element computing (Figure 4c,d) and in situ resistive strain sensor readings were highlighted at point P1 next to the tank access stairs. An approximately 30% higher strain/stress was observed due to the leg supporting the stairs. The tank access stairs are not regulated internationally (for example in EU regulatory document EN 13445), but according to the sensor measurements, they heavily influence the mechanical stress behaviors with important consequences regarding potential failures (cracks, explosions, etc.). Our research leads to the following recommendations:

(1) Regarding attached linked stairs, these must be considered in the regulatory documents for the structural behavior (such as EU regulation EN 13445).

(2) Possible solutions can be: changing the position (relocation) of the supporting structure of the access stairs in the case of the classical projects with a large number of operating hours (for example to periodically change the stairs from one position to another around the sphere tank), which is a measure that can be combined with the modification of an access stair support scheme, especially for newly designed tanks, by using technical solutions that would result in an increase of the number of degrees of freedom of the support stair scheme (for example, “joint”-type support point network).

Another conclusion is that, excluding the least stressed points (P3) and the most stressed ones (P1), the theoretical FEM results and the experimental ones obtained through the measurements usingthe rosette-type three-direction sensors were very similar. This shows once more that such sensor measurements are very reliable and useful for the mechanical behavior assessment concerning large structural tanks.

## Figures and Tables

**Figure 1 sensors-20-00525-f001:**
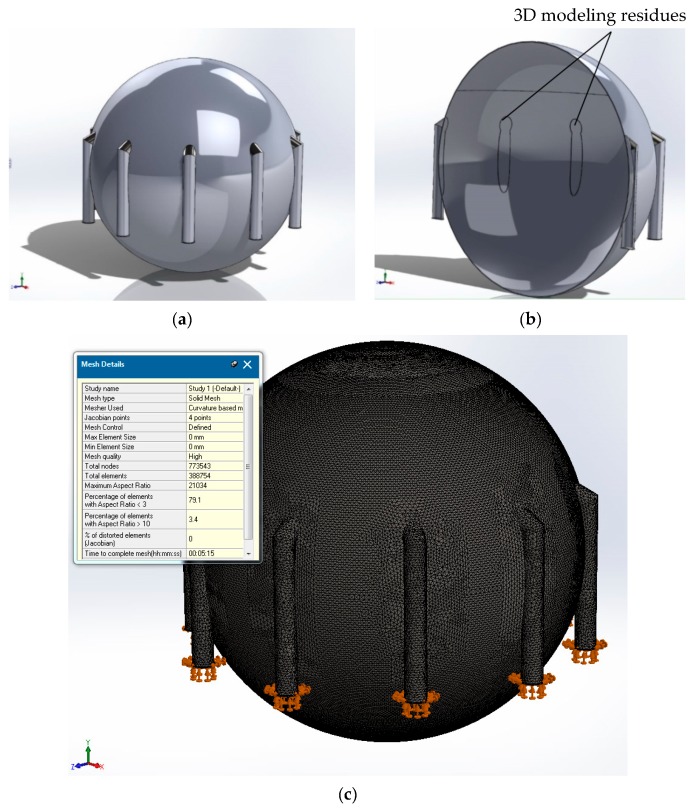
3D model of full “part” option simulation parameters: (**a**) tank simulation model using the “part” option, (**b**) tank simulation modeling issues using the “part” option, and (**c**) 3D modeling using the unique study element finite element method (FEM) meshing with 388,754 elements for a single full part case.

**Figure 2 sensors-20-00525-f002:**
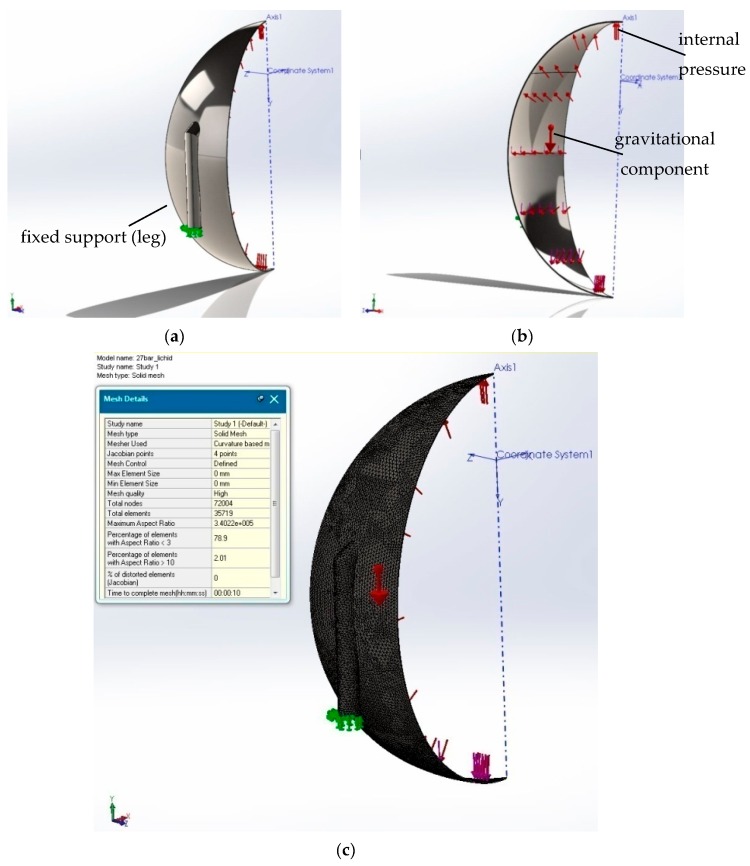
3D model of the “slice part” option simulation parameters: (**a**) SolidWorks representation rules concerning the boundary conditions in terms of the degrees of freedom, (**b**) SolidWorks representation rules concerning the boundary conditions in terms of the loading parameters, and (**c**) 3D modeling using symmetry FEM meshing with 35,719 elements for each slice part.

**Figure 3 sensors-20-00525-f003:**
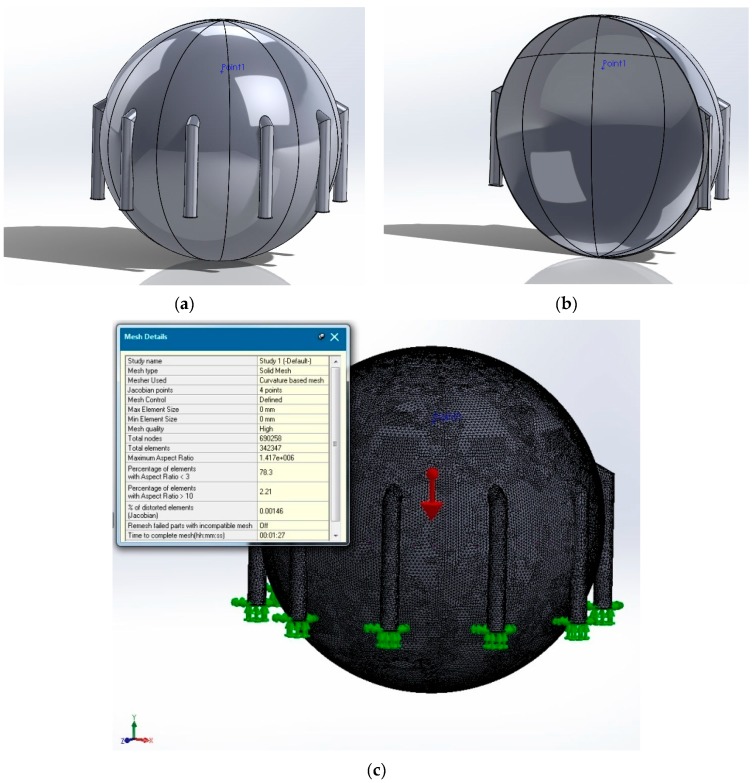
3D modeling through an “assembly” set consisting of ten slice modules: (**a**) tank simulation model using the “assembly” option, (**b**) better 3D modeling quality with the absence of issues using the “assembly” option, and (**c**) mesh elements using a ten-slice modular assembly.

**Figure 4 sensors-20-00525-f004:**
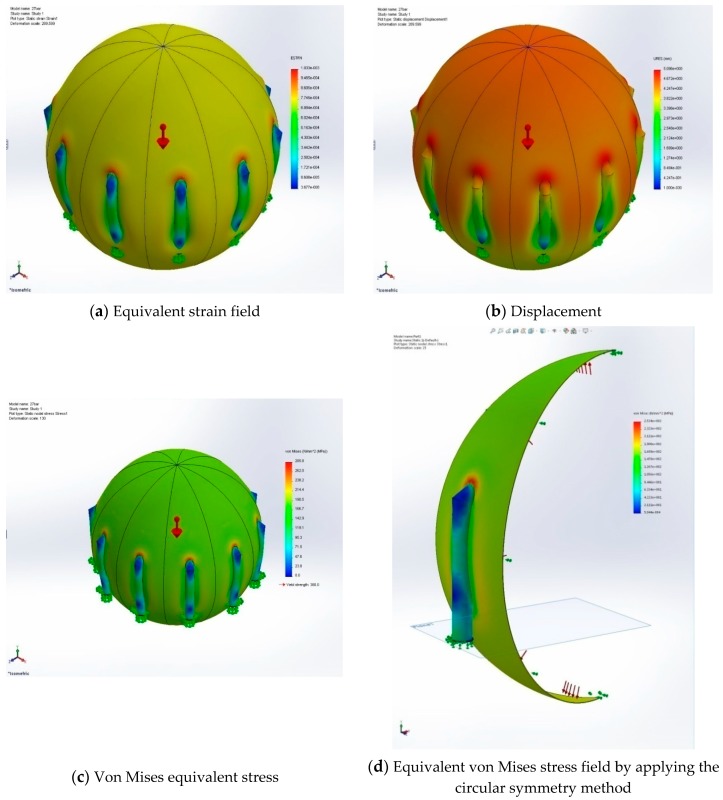
(**a**) The equivalent strain field distribution. It can be seen that the most problematic stress area of the tank (zone with color red in the figure) was the place where the supporting legs were linked to the main tank. This gives the points where the resistive sensors were placed for the experimental data acquisition. (**b**) The actual displacements of the tank (in mm) are represented. As expected, the zones of largest displacements (color red in the figure) are also (as in (**a**)) the places where the supporting legs were linked to the main tank. (**c**) The von Mises equivalent stress is given. These values were compared to the theoretical ones. In what follows we will describe the results of FEM simulations for the option “part’ with the application of the circular symmetry method. From the whole set of generated data, we present only the equivalent von Mises stress field ones (**d**), where similar conclusions can be drawn for the rest of the results. When compared with (**c**), it can be seen that the von Mises distributions were similar and the range values were equivalent. This validated the fact that using just one slice with this circular symmetry option was a sufficient replacement for the whole FEM computations.

**Figure 5 sensors-20-00525-f005:**
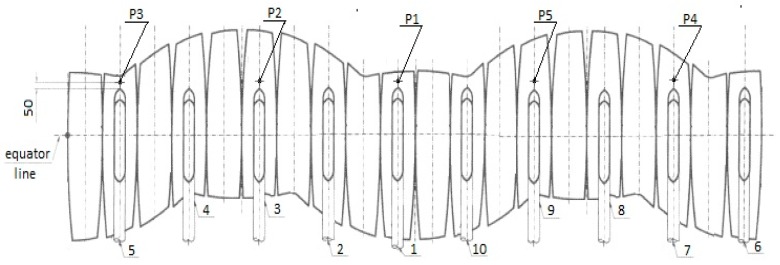
The plan for the resistive strain sensor positions.

**Figure 6 sensors-20-00525-f006:**
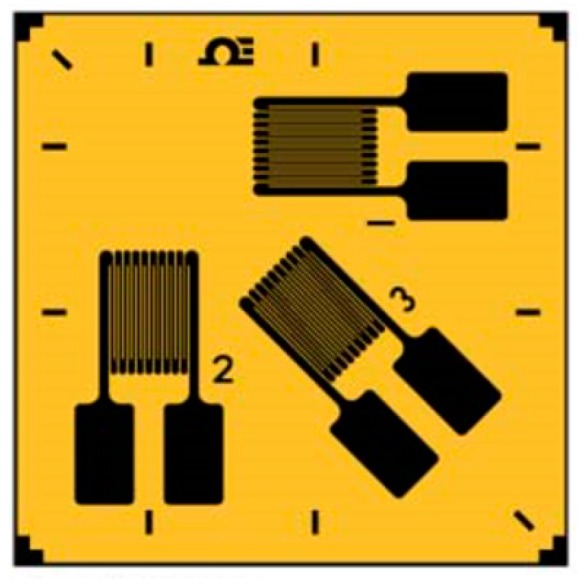
Three-directional strain sensor.

**Figure 7 sensors-20-00525-f007:**
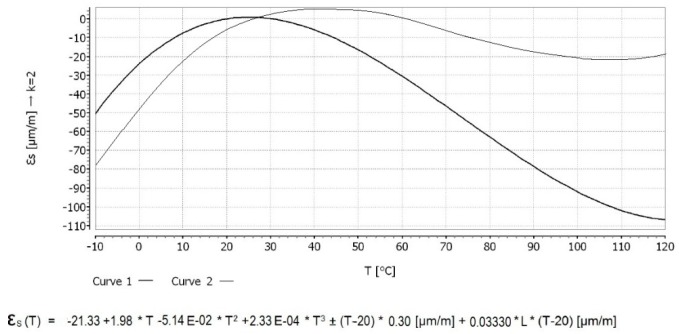
Three-directional strain sensor characteristics.

**Table 1 sensors-20-00525-t001:** Main features of the spheres that were investigated in this study. LPG: liquid petroleum gas.

No.	Volume (m^3^)	Diameter (mm)	Working Fluid	Working Pressure (MPa)	Test Pressure (MPa)	Manufacture Year
1	1800	15,100	LPG	1	1.3	1984
2	1000	12,400	LPG	1	1.3	1982
3	1000	12,400	LPG	0.5	0.6	1981
4	1000	12,400	LPG	1	1.3	1980
5	1000	12,400	LPG	0.6	0.6	1981
6	1800	15,100	LPG	1	1.3	1983
7	1000	12,400	LPG	0.6	0.6	1981
8	1000	12,400	LPG	1	1.3	1983
9	1800	15,100	LPG	1	1.3	1981
10	1800	15,100	LPG	1	1.3	1980
11	1800	15,100	LPG	2.65	2.65	1983
12	1000	12,400	LPG	1	1.3	1980
13	1000	12,400	LPG	0.8	0.8	1979
14	1000	12,400	LPG	0.8	0.8	1979
15	1000	12,400	LPG	0.8	0.8	1979
16	1000	12,400	Gasoline	2.4	3	1978
17	1000	12,400	Gasoline	2.4	3	1979
18	1000	12,400	NH_3_	2.1	2.7	1964
19	1000	12,400	NH_3_	2.1	2.7	1964

**Table 2 sensors-20-00525-t002:** Mantle material properties.

Material Properties	Value
Tensile strength at 20 °C temperature	560 MPa
Yield threshold at specified temperature	380 MPa

**Table 3 sensors-20-00525-t003:** Determined values for the most strained/stressed points.

Sphere	Maximum Stress Per Supporting Leg (MPa)					Calculated Stress	Overload Per Leg Relative to the Most Stressed Leg (in All Cases P1) %			
	P1	P2	P3	P4	P5	FEM	P2	P3	P4	P5
1	169	124	110	125	128	136	36.29	53.63	35.2	32.03
2	181	144	109	138	140	145	25.69	66.05	31.16	29.29
3	185	121	107	127	132	125	52.89	72.89	45.67	40.15
4	208	162	113	156	159	170	28.40	84.07	33.33	30.82
5	104	86	65	82	85	90	20.93	60.00	26.83	22.35
6	221	152	124	142	149	154	45.39	78.22	55.63	48.32
7	296	229	171	226	233	239	29.25	77.17	46.84	27.03
8	331	255	168	244	251	253	29.80	42.59	32.76	31.87
9	202	151	130	144	149	155	33.77	55.38	40.28	35.57
10	213	166	133	152	163	165	28.31	60.15	40.13	30.67
11	322	261	193	252	258	165	23.37	66.83	27.78	24.81
12	284	208	151	181	216	210	36.54	88.07	56.91	31.48
13	199	143	104	128	136	145	39.16	91.34	55.47	46.32
14	208	155	109	141	149	155	34.19	90.82	47.52	39.60
15	164	128	94	116	121	130	28.13	74.46	41.38	35.54
16	323	253	206	236	249	255	27.27	56.31	36.44	29.32
17	299	244	192	219	231	255	22.54	55.73	36.53	29.44
18	169	126	102	118	123	128	34.13	65.69	43.22	37.40
19	159	116	99	109	120	128	37.07	60.60	45.87	32.50

**Table 4 sensors-20-00525-t004:** The values calculated using Tukey’s test.

	P1	P2	P3	P4	P5	FEM
**Means** ${\bar{X}}_{j}$	223	169.6842	130.5263	159.7895	168	168.5789
***SE***	4368.889	2965.45	1513.152	2600.064	2853.778	2465.368
***q***	2.869446	0.065375	2.629667	0.538309	0.034601	

**Table 5 sensors-20-00525-t005:** Calculated *q_a_* values.

Threshold	10%	5%	1%
***q_a_***	1.333	1.74	2.11

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
