# Peer review of "Use of a Novel Resistive Strain Sensor Approach in an Experimental and Theoretical Study Concerning Large Spherical Storage Tank Structure Behavior During Its Operational Life and Pressure Tests"

_sensors, 2020, doi:10.3390/s20020525_

Round 1
Reviewer 1 Report
In this manuscript, large spherical storage tank structure behavior was studied. However, there are some problems in the manuscript.
Figure 5 and Figure 7 need repainting. More experimental study are necessary for evaluating the resistive strain sensor approach.I would suggest to revise the paper to make it proper representative of the presented work.
Author Response
Point 1: Figure 5 and Figure 7 need repainting.
Response 1: Figure 5 was deleted with the entire corresponding section. Figure 7 became figure 2. This figure represents a direct meshing grid capture performed by the used software. The dark aspect is due directly to the huge number of meshing elements. One of the purposes of the article is to provide a solution of such analysis on large real equipment.
Point 2: More experimental study are necessary for evaluating the resistive strain sensor approach
Response 2: The results synthesized and presented in table 2 of the paper represent the result of more than 8 years of measurements made on this type of equipment. The large spherical storage pressure vessels are critical equipment with limited access. The authors had the opportunity to investigate and make measurements on 19 such equipments. Furthermore, the present article relates to in situ performed measurements and not classic laboratory strain sensor approach. One of the purposes of the article is to optimize the use of resistive sensors. This optimization focuses on identifying areas of interest and the minimum number of measurement points required. We consider that the obtained and reported results are significant due to the fact that the authors of the paper, who carried out the measurements, are officially authorized by the National Authority to carry out such investigations and issue examination bulletins for all types of pressure vessels with volumes over 100 cubic meters.
Point 3: I would suggest to revise the paper to make it proper representative of the presented work.
Response 3: The paper was revised according to all three reviewer comments.

Reviewer 2 Report
This work reports on the implementation of strain sensors to monitor spherical storage tank structure behaviour during operation and pressure test. While there have been a number of studies on using resistive strain sensors in storage tank; these works reported in the literature only deal with small model, with dimensions much smaller that the real applications. The present work offers novelty regarding evaluation of the FEM for analytical stimulation and practical measurement with large-scale system (e.g. with the volume of up to 1800 cubic meters.)
However, the present form of the manuscript is not well organized. The writing is very poor and unclear. The introduction is relatively long and not direct to the point. Therefore, major revision is required before further consideration of publication in Sensors.
Please address the following concerns:
The abstract is too general. The outline of the paper should not be mentioned in the abstract; instead the author should present the quantitative results found in this work and emphasize their novelty. The introduction is not easy to follow and there are too many small paragraphs that do not make a clear connection. The reviewer suggests the authors to restructure this section. For instance, the introduction should provide the background of this field, work have been done in the literature (including finite element method, and experimental studies on small scale model). Then the authors should present the evidence of using small size model in the test, any paradox between FEM and experimental data … And finally, the proposed approach that will be described in the following section. The material properties of the spherical tank (lines 133 to 134) should be listed in a table format. Again, section 2 (material and methods consist of too many small paragraphs. Should start with: (i) materials and then (ii) empirical approach (or laboratory application) and finite element method). This paragraph seems unnecessary. “Under these circumstances, the use of experimental methods for determining the state of tension and deformation in strained bodies becomes a mandatory condition. Most often, experimental methods are used in parallel to analytical ones. In fact, experimental methods are based on theoretical knowledge and the conclusions drawn from it. Both methods have advantages and disadvantages. Applied together, the two methods lead to very good results. Experimental results can confirm or invalidate the analytically obtained results.” Again, this paragraph is not necessary “FEM analysis has been performed using a commercial software solution – Dassault Systèmes SolidWorks1 – discussed in full in the next Section; the Finite Element Method (FEM) is “one of the most powerful engineering methods for numerical calculation in use today. The instrument becomes even more powerful as this method has been attached to some design programs thus obtaining possibilities to simulate the behavior of the equipment in operation both from a static and dynamic point of view (SolidWorks Simulation user manual)”. Adding this non-sense paragraph (without new insight) in fact brings down the quality of the manuscript. “These points are selected to give optimal numerical results. The program calculates stresses at the nodes of each element by extrapolating the results available at the Gaussian points. After a successful run, nodal stress results at each node of every element are available in the database” (SolidWorks Simulation user manual).” Should be removed. The parameters listed in Figure 6 and 7 are not clear. The contents with equations shown in pages 5,6,7 are just from text book. Since this work employed FEM, but not numerical calculation. The present of these equations seem unnecessary. Figure 9 should show the FEM result only but not the screen short of the software interface! Figures 9, 10, 11 should be combined into a single figure with caption “FEM results. (a) Equivalent strain field; (b) Displacement components; (c)von Mises equivalent stress.” The strain sensors have three components forming a rosette structure. Please provide equation to drive the stress from the measured strain. What are the curve 1 and curve 2 in Figure 16. The results are not well represented. Please add the raw data of the sensors. There are many typos and grammatical errors in the manuscript. For instance, this sentence does not have a verb. “Due to strain field presence the element deforms, resulting in normal, tangential stresses (σ, τ), experimental direct determination of such tensile parameters being impossible.”

Author Response
Point 1: The abstract is too general. The outline of the paper should not be mentioned in the abstract; instead the author should present the quantitative results found in this work and emphasize their novelty.
Response 1: The abstract was revised according to the reviewer comments.
Point 2: The introduction is not easy to follow and there are too many small paragraphs that do not make a clear connection. The reviewer suggests the authors to restructure this section. For instance, the introduction should provide the background of this field, work have been done in the literature (including finite element method, and experimental studies on small scale model). Then the authors should present the evidence of using small size model in the test, any paradox between FEM and experimental data …
Response 2: The introduction was revised according to the reviewer comments.
Point 3: And finally, the proposed approach that will be described in the following section. The material properties of the spherical tank (lines 133 to 134) should be listed in a table format. Again, section 2 (material and methods consist of too many small paragraphs. Should start with: (i) materials and then (ii) empirical approach (or laboratory application) and finite element method). This paragraph seems unnecessary. “Under these circumstances, the use of experimental methods for determining the state of tension and deformation in strained bodies becomes a mandatory condition. Most often, experimental methods are used in parallel to analytical ones. In fact, experimental methods are based on theoretical knowledge and the conclusions drawn from it. Both methods have advantages and disadvantages. Applied together, the two methods lead to very good results. Experimental results can confirm or invalidate the analytically obtained results.” Again, this paragraph is not necessary “FEM analysis has been performed using a commercial software solution – Dassault Systèmes SolidWorks1 – discussed in full in the next Section; the Finite Element Method (FEM) is “one of the most powerful engineering methods for numerical calculation in use today. The instrument becomes even more powerful as this method has been attached to some design programs thus obtaining possibilities to simulate the behavior of the equipment in operation both from a static and dynamic point of view (SolidWorks Simulation user manual)”. Adding this non-sense paragraph (without new insight) in fact brings down the quality of the manuscript. “These points are selected to give optimal numerical results. The program calculates stresses at the nodes of each element by extrapolating the results available at the Gaussian points. After a successful run, nodal stress results at each node of every element are available in the database” (SolidWorks Simulation user manual).” Should be removed.
Response 3: The paragraphs deemed unnecessary or meaningless were eliminated. The section has been restructured according to the comments.
Point 4: The parameters listed in Figure 6 and 7 are not clear.
Response 4: Figures 6, 7 became figures 1, 2. This figures represents a direct meshing grid capture performed by the used software. The dark aspect is due directly to the huge number of meshing elements. One of the purposes of the article is to provide a solution of such analysis on large real equipment.
Point 5: The contents with equations shown in pages 5,6,7 are just from text book. Since this work employed FEM, but not numerical calculation. The present of these equations seem unnecessary.
Response 5: The content was completely eliminated.
Point 6: Figure 9 should show the FEM result only but not the screen short of the software interface! Figures 9, 10, 11 should be combined into a single figure with caption “FEM results. (a) Equivalent strain field; (b) Displacement components; (c)von Mises equivalent stress.” The strain sensors have three components forming a rosette structure.
Response 6: The issue has been addressed.
Point 7: Please provide equation to drive the stress from the measured strain. What are the curve 1 and curve 2 in Figure 16. The results are not well represented. Please add the raw data of the sensors.
Response 7: Stress-strain relation is based on generic Hooke’s law, widely used in Mechanics of Materials area. Figure 16 became figure 6 and the desired information is now included. An appendix containing the raw data and auxiliary information was generated. The authors of the paper, who carried out the measurements, are officially authorized by the National Authority to carry out such investigations and issue examination bulletins for all types of pressure vessels with volumes over 100 cubic meters.
Point 8: There are many typos and grammatical errors in the manuscript. For instance, this sentence does not have a verb. “Due to strain field presence the element deforms, resulting in normal, tangential stresses (σ, τ), experimental direct determination of such tensile parameters being impossible.”
Response 8: The issue has been addressed.

Reviewer 3 Report
Referee Report on:
“Use of novel resistive strain sensor approach in experimental and theoretical study concerning large spherical storage tank structure behavior during operational life and pressure tests”
The manuscripts could be interesting, however major issues rose in my opinion. Firstly authors should better state their contributions and completely avoid the inclusion of parts taken from other publications. Secondly, the obtained results cannot be considered as general and valid, because they are not obtained by applying (to the experimental data) suitable methods, such as statistical modelling (well known in literature).
Major comments
Introduction and Section 2In my opinion in this section, the authors should describe more in detail and stress better the main contributions of the manuscript. For instance, in Section 2.2, p. 11, lines 287-289 the authors state that “The originality of this work pertains to the use of slices for the FEM 3D modeling and simulations for large structures – experiments with large structures. This slice modeling technique has never and nowhere been reported in the literature, to the best of our knowledge.” This point should be also anticipated in the Introduction.
Moreover, the Introduction should be reorganized so that first, the literature review should be described (e.g. p. 2, lines 71-75 and lines 80-93), and then the main contributions of the manuscript should be clearly stated (e.g. p. 2, lines 67-70 and lines 75-79).
The Finite Element Method is the core method considered in the manuscript. Therefore, when the authors refer to this method, they should cite the most relevant literature related to it, and not include entire sentences taken from the CosmosWorks/SolidWorks Simulation user manual, as at p. 2, lines 60-66.
Materials and Methods at p. 4, lines 135-136, the authors state that “Section 2.3 provides the description of the theoretical experiment.” There is no Section 2.3 in the Manuscript. Do the author refer maybe to Section 3.2 although it is related to the Results? This point should be made clear. More precisely, if the authors refer to Section 3.2., this section should be anticipated in Materials and Methods. Otherwise, the author should include this missing section. Section 2.1: Analytical MethodAll the symbols used to describe the theory on the analytical method should be defined. More precisely, p. 5 , line 163, what the symbol “E” stands for? Moreover, at p. 7, line 184, please explain what , , and represent in Equation (1). Lastly, the most relevant literature on the analytical method should be cited and not inserted (through entire sentences taken from the CosmosWorks/SolidWorks Simulation 2014 Static) directly in the text.
Page 11, lines 287-289: the authors state that ““The originality of this work pertains to the use of slices for the FEM 3D modeling and simulations for large structures – experiments with large structures. This slice modeling technique has never and nowhere been reported in the literature, to the best of our knowledge.” It is not clear for me how the slices are introduced in the FEM modelling. The authors should explicitly report how the FEM modelling theory when slices are considered. Results and Discussions Section 3.3My main concern is the validity of some of the results reported in Section 3.3. According to the authors this is one of the main contribution of the manuscript. In fact, in Section 2 at page 2, lines 67-69, they state that: “The present paper proposes a comparative study of the results obtained by applying the Finite Element Method (FEM) and the results obtained following resistive strain sensor measurements during in situ overpressure tests.” More precisely, according to Section 3.2, first, FEM simulations are performed and the zones in which the resistive strain sensors are actually placed are determined. Afterwards, using sensors mounted in these places, the authors obtained the measurements on the experimental data for the 19 spheres, also reported in Table 2. According to the description, it seems like the “Maximum stress per supporting leg” reported for the 19 spheres is the output of the experiment, i.e. a response variable. Therefore, the comparison of these values with the results obtained through the FEM modelling is purely descriptive, and based on quantitative indexes/measurements. In my opinion, in order to conclude that these results are really valid, the authors should consider suitable statistical models for the experimental data, and not only a simple comparison among the obtained measurements.
Minor Comments:
Page 1, line 10: there is a missing part in the email address “mocanustef@gmail”. Page 1, lines 38-39: please, cite to which existing literature you refer to. Page 2, line 73: insert the number for the reference “Agbo et al.” . Page 3, lines 102-103: please, add one or two sentences in order to properly introduce the problem under study. Page 4, line 140: there is a missing full stop at the end of the sentence. Page7, line 199 and Figure 5: please use only one “Gaussian points” or “Gauss points”. Page 8, line 223: correct “One” in “one”. Page 8, line 225: correct “Second” in “second”. Page 10, line 278: correct “approx.” in “approximately”. Page 14, line 340: it seems that “(Section 4)” is incorrect since it relates to the Conclusion. Page 16, line 395: “From the figure,…” Which figure? Do you maybe mean Table 2? Page 17, line 422: incorrect numbering of the Section. It should be “4. Conclusions” and not “2. Conclusions”.Author Response
Point 1: The manuscripts could be interesting, however major issues rose in my opinion. Firstly authors should better state their contributions and completely avoid the inclusion of parts taken from other publications. Secondly, the obtained results cannot be considered as general and valid, because they are not obtained by applying (to the experimental data) suitable methods, such as statistical modelling (well known in literature).
Response 1: The manuscript has been restructured. For validating the results a statistically model has been used (Kolmorogov-Smirnov test).
Point 2: Introduction and Section 2
In my opinion in this section, the authors should describe more in detail and stress better the main contributions of the manuscript. For instance, in Section 2.2, p. 11, lines 287-289 the authors state that “The originality of this work pertains to the use of slices for the FEM 3D modeling and simulations for large structures – experiments with large structures. This slice modeling technique has never and nowhere been reported in the literature, to the best of our knowledge.” This point should be also anticipated in the Introduction.
Moreover, the Introduction should be reorganized so that first, the literature review should be described (e.g. p. 2, lines 71-75 and lines 80-93), and then the main contributions of the manuscript should be clearly stated (e.g. p. 2, lines 67-70 and lines 75-79).
The Finite Element Method is the core method considered in the manuscript. Therefore, when the authors refer to this method, they should cite the most relevant literature related to it, and not include entire sentences taken from the CosmosWorks/SolidWorks Simulation user manual, as at p. 2, lines 60-66. Materials and Methods at p. 4, lines 135-136, the authors state that “Section 2.3 provides the description of the theoretical experiment.” There is no Section 2.3 in the Manuscript. Do the author refer maybe to Section 3.2 although it is related to the Results? This point should be made clear. More precisely, if the authors refer to Section 3.2., this section should be anticipated in Materials and Methods. Otherwise, the author should include this missing section. Section 2.1: Analytical Method
All the symbols used to describe the theory on the analytical method should be defined. More precisely, p. 5 , line 163, what the symbol “E” stands for? Moreover, at p. 7, line 184, please explain what , , and represent in Equation (1). Lastly, the most relevant literature on the analytical method should be cited and not inserted (through entire sentences taken from the CosmosWorks/SolidWorks Simulation 2014 Static) directly in the text.
Response 2: The introduction and the mentioned section were revised according to the all reviewers comments. The paragraphs deemed unnecessary or meaningless were eliminated. The sections were renumbered according to eliminated content.
Point 3: Page 11, lines 287-289: the authors state that ““The originality of this work pertains to the use of slices for the FEM 3D modeling and simulations for large structures – experiments with large structures. This slice modeling technique has never and nowhere been reported in the literature, to the best of our knowledge.” It is not clear for me how the slices are introduced in the FEM modelling. The authors should explicitly report how the FEM modelling theory when slices are considered. Results and Discussions Section 3.3
Response 3: The issue has been addressed according to requests. It is of major interest the fact that internal questions concerning the software structure and numerical calculation algorithm are not directly accessible to authors and protected by global copywrite laws.
Point 4: My main concern is the validity of some of the results reported in Section 3.3. According to the authors this is one of the main contribution of the manuscript. In fact, in Section 2 at page 2, lines 67-69, they state that: “The present paper proposes a comparative study of the results obtained by applying the Finite Element Method (FEM) and the results obtained following resistive strain sensor measurements during in situ overpressure tests.” More precisely, according to Section 3.2, first, FEM simulations are performed and the zones in which the resistive strain sensors are actually placed are determined. Afterwards, using sensors mounted in these places, the authors obtained the measurements on the experimental data for the 19 spheres, also reported in Table 2. According to the description, it seems like the “Maximum stress per supporting leg” reported for the 19 spheres is the output of the experiment, i.e. a response variable. Therefore, the comparison of these values with the results obtained through the FEM modelling is purely descriptive, and based on quantitative indexes/measurements. In my opinion, in order to conclude that these results are really valid, the authors should consider suitable statistical models for the experimental data, and not only a simple comparison among the obtained measurements.
Response 4: As mentioned above, for validating the results a statistically model has been used (Kolmorogov-Smirnov test). The results synthesized and presented in table 2 of the paper represent the result of more than 8 years of measurements made on this type of equipment. The large spherical storage pressure vessels are critical equipment with limited access. The authors had the opportunity to investigate and make measurements on 19 such equipments. Furthermore, the present article relates to in situ performed measurements and not classic laboratory strain sensor approach. One of the purposes of the article is to optimize the use of resistive sensors. This optimization focuses on identifying areas of interest and the minimum number of measurement points required. We consider that the obtained and reported results are significant due to the fact that the authors of the paper, who carried out the measurements, are officially authorized by the National Authority to carry out such investigations and issue examination bulletins for all types of pressure vessels with volumes over 100 cubic meters.
Point 5: Minor Comments:
Page 1, line 10: there is a missing part in the email address “mocanustef@gmail”. Page 1, lines 38-39: please, cite to which existing literature you refer to. Page 2, line 73: insert the number for the reference “Agbo et al.” . Page 3, lines 102-103: please, add one or two sentences in order to properly introduce the problem under study. Page 4, line 140: there is a missing full stop at the end of the sentence. Page7, line 199 and Figure 5: please use only one “Gaussian points” or “Gauss points”. Page 8, line 223: correct “One” in “one”. Page 8, line 225: correct “Second” in “second”. Page 10, line 278: correct “approx.” in “approximately”. Page 14, line 340: it seems that “(Section 4)” is incorrect since it relates to the Conclusion. Page 16, line 395: “From the figure,…” Which figure? Do you maybe mean Table 2? Page 17, line 422: incorrect numbering of the Section. It should be “4. Conclusions” and not “2. Conclusions”.
Response 5: All issues have been addressed.

Round 2
Reviewer 1 Report
In this manuscript, large spherical storage tank structure behavior was studied. However, there are some problems in the manuscript.
Please check the manuscript for linguistic errors (e.g. fem in Line 293). Tables should be noted for Line 295-298.I would suggest to revise the paper to make it proper representative of the presented work.
Author Response
We are very grateful to the reviewers for their critical comments and thoughtful suggestions. Based on these comments and suggestions, we have made careful modifications to the original manuscript. The point-to-point replies and explanations for all of the revisions are listed below for easy reference.
Point 1: Please check the manuscript for linguistic errors (e.g. fem in Line 293). Tables should be noted for Line 295-298.
I would suggest to revise the paper to make it proper representative of the presented work.
Response 1: We fully agree with your observation. The changes according to the requirements of all reviewers have been made. As a result, the paper has been revised. Furthermore the paper was reviewed by two native English speakers in order to address language related issues.

Reviewer 3 Report
Referee Report#2 of the revised manuscript:
“Use of novel resistive strain sensor approach in experimental and theoretical study concerning large spherical storage tank structure behaviour during operational life and pressure tests”
Major comments
IntroductionAs suggested in my previous revision report, the authors have reorganized the introduction section so that i) the literature review is illustrated, and following, the main contribution of the manuscript is described. However, two key-points related to the Introduction and included in my previous report are not addressed; more precisely:
the authors should cite the most relevant literature when referring to the Finite Element Method (FEM); in this version of the manuscript there is not any relevant reference related to the FEM; in the previous version of the manuscript the authors claimed that “The originality of this work pertains to the use of slices for the FEM 3D modeling and simulations for large structures – experiments with large structures. This slice modeling technique has never and nowhere been reported in the literature, to the best of our knowledge.” In this version of the manuscript, this sentence has been modified as follows: “The originality of this work is given by the use of SolidWorks symmetry algorithm for the FEM 3D modeling and simulations, for large structures. To the best of our knowledge for complex and large structures this slicing modeling technique has never and nowhere been reported in the literature.” Why this point is not explicitly highlighted also in the main contribution of the manuscript in the Introduction, as suggested in my previous report? Materials and Methods I requested to the authors to properly define all the symbols used to describe the theory on the Finite Element Method, and to also illustrate how the slices the authors applied are introduced in the FEM modelling. In this version of the manuscript, these issues are not addressed, but actually, the section on theory has been completely deleted. Why? In my opinion, this Section theory is necessary; otherwise it is extremely difficult to understand and evaluate the manuscript; at the end of Section 2 the authors write the following sentence: “The authors' initial purpose was to compare the experimental results measured with the simulated values in the FEM. In fact, unexpected results were obtained.” How does this sentence refer to the previous version of the manuscript, and to which “unexpected results” the authors refer to? Please, be more clear. Results and Discussions: Section 3.3 My main concern is again about the validity of some of the results reported in Section 3.3, a key-point that is not properly addressed by the authors. First, the authors state that “for validating the results a statistically model has been used (Kolmorogov-Smirnov test)”. This statement is completely wrong: the Kolmogorov-Smirnov test is not a statistical model but a not parametric hypothesis test. Moreover, the results obtained through the Kolmogorov-Smirnov test could be considered only as an initial exploratory analysis for the data under study. This implies that, in order to state that the results are really valid, suitable statistical models should be applied. Secondly, the following statement the authors report in the cover letter “We consider that the obtained and reported results are significant due to the fact that the authors of the paper, who carried out the measurements, are officially authorized by the National Authority to carry out such investigations and issue examination bulletins for all types of pressure vessels with volumes over 100 cubic meters” has absolutely no relevance for the statistical validity of the results. Furthermore, the following sentence (at p. 13, line 289) “For statistical validation, the normality Kolmogorov-Smirnov Test (KS test) was used, …” is not accurate; what the author mean with “the normality Kolmogorov-Smirnov test”?
Minor Comments:
Page 1, lines 38-39: as requested in my previous report, please, cite to which existing literature you refer to. This point is not addressed. the acronym “FEM” is defined twice in the paper, i.e. p. 3 (lines 79-80) and p. 3 (line 94). Please, define it just one time, e.g. at p. 1, line31, and following use only the acronym. 2, line 40: the sentence “In accordance with the legislation, …”, which legislation? Please add the appropriate reference. 5, line 136: correct “fem” in “FEM”. 13, line 288: the correct name is “Kolmogorov-Smirnov test” and not “Kolmorogov-Smirnov test” 13, line 292: correct “An comparative study…” to “A comparative study…”Author Response
We are very grateful to the reviewers for their critical comments and thoughtful suggestions. Based on these comments and suggestions, we have made careful modifications to the original manuscript. The point-to-point replies and explanations for all of the revisions are listed below for easy reference. We hope that we have understood this time what changes are needed to increase the level of the paper.
Point 1: The authors should cite the most relevant literature when referring to the Finite Element Method (FEM); in this version of the manuscript there is not any relevant reference related to the FEM;
Response 1: We thank the reviewer for the relevant observation. We consider SolidWorks Dassault theoretical background (handbook) as relevant for the FEM domain. Furthermore we added two extra references related to FEM – SolidWorks correlation that we consider relevant, respectively [22,23].
Point 2: in the previous version of the manuscript the authors claimed that “The originality of this work pertains to the use of slices for the FEM 3D modelling and simulations for large structures – experiments with large structures. This slice modelling technique has never been reported in the literature, to the best of our knowledge.” In this version of the manuscript, this sentence has been modified as follows: “The originality of this work is given by the use of SolidWorks symmetry algorithm for the FEM 3D modelling and simulations, for large structures. To the best of our knowledge for complex and large structures this slicing modelling technique has never and nowhere been reported in the literature.” Why this point is not explicitly highlighted also in the main contribution of the manuscript in the Introduction, as suggested in my previous report?
Response 2: We fully agree with your observation, therefore the necessary information has been added in the Introduction section. (Lines 85, 86)
Point 3: Materials and Methods I requested to the authors to properly define all the symbols used to describe the theory on the Finite Element Method, and to also illustrate how the slices the authors applied are introduced in the FEM modelling. In this version of the manuscript, these issues are not addressed, but actually, the section on theory has been completely deleted. Why? In my opinion, this Section theory is necessary; otherwise it is extremely difficult to understand and evaluate the manuscript;
Response 3: We thank the reviewer for the relevant observation. Initially we introduced in-extenso this part, which referred to the basic elements of the Theory of Elasticity with direct application on the FEM calculus; we also considered it useful to create an overview on the mathematical apparatus underlying the solver implicitly used. The decision to remove this part from the work was based on the comments from 25.11.2019 of another reviewer who said "The contents with equations shown in pages 5,6,7 are just from text book. Since this work employed FEM, but not numerical calculation. The present of these equations seem unnecessary." For that reason, we deleted this section.
Because we consider your comment important we added Appendix B entitled “FEM computational approach – Theory of Elasticity basic knowledge”.
Point 4: at the end of Section 2 the authors write the following sentence: “The authors' initial purpose was to compare the experimental results measured with the simulated values in the FEM. In fact, unexpected results were obtained.” How does this sentence refer to the previous version of the manuscript, and to which “unexpected results” the authors refer to? Please, be clearer.
Response 4: The unexpected character of the conclusions from a technical point of view is related to the major counterintuitive influence of the access stairs on the mechanical behaviour of the tanks. This fact is evidenced by the difference between the values measured with the help of tensometric sensors in all 19 spheres investigated in comparison with the values calculated by FEM. No regulations at national or international level are taking into account stairs effect. To the best of our knowledge, nowhere in the literature is reported about this result. According to the results, our recommendation is that access stairs should be taken into account by existing regulation related to the mechanical behavior of the tanks. Those comments are also added into the paper.
Point 5: Results and Discussions: Section 3.3 My main concern is again about the validity of some of the results reported in Section 3.3, a key-point that is not properly addressed by the authors. First, the authors state that “for validating the results a statistically model has been used (Kolmorogov-Smirnov test)”. This statement is completely wrong: the Kolmogorov-Smirnov test is not a statistical model but a not parametric hypothesis test. Moreover, the results obtained through the Kolmogorov-Smirnov test could be considered only as an initial exploratory analysis for the data under study. This implies that, in order to state that the results are really valid, suitable statistical models should be applied. Secondly, the following statement the authors report in the cover letter “We consider that the obtained and reported results are significant due to the fact that the authors of the paper, who carried out the measurements, are officially authorized by the National Authority to carry out such investigations and issue examination bulletins for all types of pressure vessels with volumes over 100 cubic meters” has absolutely no relevance for the statistical validity of the results. Furthermore, the following sentence (at p. 13, line 289) “For statistical validation, the normality Kolmogorov-Smirnov Test (KS test) was used, …” is not accurate; what the author means with “the normality Kolmogorov-Smirnov test”?
Response 5: We very much appreciate the reviewer’s detailed evaluations and suggestions; therefore, the manuscript has been revised thoroughly according to the reviewer’s advice. With the kind support of professor Mazilu from the Department of Mathematics and Statistics, it was decided to replace the Kolmogorov-Smirnov test with the Tukey test. Through this test, which verifies the equality of the means for k selections of possibly different volumes, we aimed to identify the similarities and differences between the values measured experimentally for each supporting leg and the value calculated using FEM.
Point 6: Minor Comments:
Page 1, lines 38-39: as requested in my previous report, please, cite to which existing literature you refer to. This point is not addressed. the acronym “FEM” is defined twice in the paper, i.e. p. 3 (lines 79-80) and p. 3 (line 94). Please, define it just one time, e.g. at p. 1, line31, and following use only the acronym. 2, line 40: the sentence “In accordance with the legislation, …”, which legislation? Please add the appropriate reference. 5, line 136: correct “fem” in “FEM”. 13, line 288: the correct name is “Kolmogorov-Smirnov test” and not “Kolmorogov-Smirnov test” 13, line 292: correct “An comparative study…” to “A comparative study…”
Response 6: All issues have been addressed.
Concerning the legislation detail issue:
For the safe operation of such equipment, there are generally regulations that are adopted by each country separately. In the case of this paper, the European regulation has been used as a reference. (The Pressure Equipment Directive (PED) (2014/68/EU); Guidelines - pressure equipment directive 97/23/EC - related to the former PED; Guidelines - pressure equipment directive 2014/68/EU)
